# Development of a Simple DNA Extraction Method and *Candida* Pan Loop-Mediated Isothermal Amplification Assay for Diagnosis of Candidemia

**DOI:** 10.3390/pathogens11020111

**Published:** 2022-01-18

**Authors:** Da Hye Lim, Hyunseul Jee, Kyung Chul Moon, Chae Seung Lim, Woong Sik Jang

**Affiliations:** 1Departments of Laboratory Medicine, College of Medicine, Korea University Guro Hospital, Seoul 08308, Korea; ldh9692@korea.ac.kr (D.H.L.); malarim@korea.ac.kr (C.S.L.); 2BK21 Graduate Program, Department of Biomedical Sciences, College of Medicine, Korea University, Seoul 08308, Korea; jhs603@korea.ac.kr; 3Emergency Medicine, College of Medicine, Korea University Guro Hospital, Seoul 08308, Korea; cmooner@korea.ac.kr

**Keywords:** *Candida* spp., *Candida albicans*, DNA extraction, Chelex-100, multiplex LAMP

## Abstract

To reduce the morbidity and mortality of candidemia patients through rapid treatment, the development of a simple, rapid molecular diagnostic method that is based on nucleic acid extraction and is superior to conventional methods for detecting *Candida* in the blood is necessary. We developed a multiplex *Candida* Pan/internal control (IC) loop-mediated isothermal amplification (LAMP) assay and a simple DNA extraction boiling protocol using Chelex-100 that could extract yeast DNA in blood within 20 min. The Chelex-100/boiling method for DNA extraction showed comparable efficiency to that of the commercial QIAamp UCP Pathogen Mini Kit using *Candida albicans* qPCR. In addition, the *Candida* Pan/IC LAMP assay showed superior sensitivity to that of general *Candida* Pan and species qPCRs against clinical DNA samples extracted with the QIAamp UCP Pathogen Mini Kit and Chelex-100/boiling method. The *Candida* Pan/IC LAMP assay followed by Chelex-100/boiling-mediated DNA extraction showed high sensitivity (100%) and specificity (100%) against clinical samples infected with *Candida*. These results suggest that the *Candida* Pan/IC LAMP assay could be used as a rapid molecular diagnostic test for candidemia.

## 1. Introduction

*Candida* species are among the top five infectious bloodstream pathogens and remain the most common cause of invasive fungal infections [1]. Invasive candidiasis, which occurs when Candida spp. infect other tissues and organs, infects more than 250,000 people worldwide each year and causes more than 50,000 deaths [2,3]. The reported candidemia mortality ranges from 30% to 60% with up to 30 days of increase in the length of hospital stay for survivors [4].

Early diagnosis is critical for appropriate patient management and for improving the outcomes of candidemia. Blood cultures, the current diagnostic gold standard, are limited by low sensitivity, ranging from 21% to 71% [5], and a slow turnaround, usually exceeding 48 h [6,7,8]. Therefore, various non-culture-based diagnostic methods such as immunoassay (mannan, anti-mannan antibodies and (1-3)-β-d-glucan (BDG) assay) [9,10,11] and PCR [8] have been developed; however, detection methods using mannan, anti-mannan antibodies and BDG have been reported to have low specificity [10,12] and PCR-based diagnosis is time-consuming, although the detection specificity is high. Therefore, the development of a fast diagnosis system is required for the detection of fungi in blood.

Recently, several isothermal amplification techniques for Candida have been proposed as molecular diagnostic methods to overcome these limitations, including loop-mediated isothermal amplification (LAMP) [13], nucleic acid sequence-based amplification (NASBA) [14] and rolling circle amplification [15]. Among these isothermal amplification methods, LAMP is the most extensively investigated method for Candida detection [13,16,17]. Inàcio et al. reported a LAMP technique for the amplification of the 26S rRNA gene in clinically relevant Candida yeasts [13]. Fallahi et al. developed a C. albicans-specific LAMP assay using fluorescence detection [16]. Hongling et al. established multiple pathogen loop-mediated isothermal amplification (LAMP) using microfluidic chip technology for Staphylococcus aureus, Escherichia coli, Pneumoniae Klebsiella, Shigella, methicillin-resistant Staphylococcus aureus (MRSA) and C. albicans [17]. LAMP is composed of six primers, of which four primers contain six parts of the target gene sequence and two loop primers react with the target gene to form a loop structure and then robustly amplify the target gene at 58–65 °C using Bst or Bsm polymerases with DNA-strand displacement activity [18,19].

To shorten the time for diagnosis, fast nucleic acid extraction from Candida in blood is needed. The sensitivity of any molecular diagnostic method for the detection of fungal pathogens depends on the lysis efficiency of fungal cells from blood samples and purification of DNA without PCR inhibitors [20]. In particular, the breakdown of the fungal cell wall is a crucial step for lysis of the cell entity and isolation of genomic DNA. Current fungal DNA extraction protocols involve enzymatic [21,22], chemical or physical disruption steps [23,24], bead beating using glass [25], or ceramic beads [26] to disrupt the fungal cell wall. Unlike nucleic acid extraction of animal cells or viruses, the additional cell wall disruption step makes rapid nucleic acid extraction from fungi more difficult.

In this study, we developed a Chelex-100/boiling DNA extraction method (within 20 min) and Candida Pan/IC LAMP assay (40 min) for the rapid diagnosis of candidemia. The performance of the Chelex-100 DNA extraction method was compared and evaluated with that of the QIAamp UCP Pathogen Mini Kit using general Candida qPCR. In addition, the sensitivity and specificity of the Candida Pan/IC LAMP assay were compared with those of two general qPCRs (Candida Pan and Candida species) for two kinds of clinical sample DNAs, which were extracted using the QIAamp UCP Pathogen Mini Kit and Chelex-100/boiling DNA extraction.

## 2. Materials and Methods

### 2.1. Clinical Samples

A total of 136 clinical whole blood samples were collected from Candida-infected (n = 36) and non-infected patients (normal control, n = 100) at Korea University Guro Hospital from January 2019 to August 2021. All clinical samples were confirmed by VITEK 2 COMPACT system (bioMérieux, Durham, NC, USA) using a VITEK^®^2 YST ID card (bioMérieux, Durham, NC, USA). True positives included clinical blood samples infected with C. albicans (n = 9), C. glabrata (n = 9), C. tropicalis (n = 9) and C. parapsilosis (n = 9). For the cross-reactivity test, the cultured bacteria samples, including Escherichia coli, Enterococcus faecium, Klebsiella spp., Staphylococcus aureus and Staphylococcus epidermidis, were obtained from the Korea University Guro Hospital. The study was conducted in accordance with the guidelines of the Declaration of Helsinki and was approved by the Institutional Review Board of Korea University Guro Hospital (2020GR0512).

### 2.2. Isolation of Genomic DNA from Candida Strains

Candida albicans (CCARM 14029), C. krusei (CCARM 14017), C. tropicalis (CCARM14019), C. parapsilosis (CCARM14016), C. auris (KCTC17850) and C. glabrata (KCTC 7219) were obtained from the Culture Collection of Antimicrobial-Resistant Microbes (CCARM; Seoul, Korea) and Korean Collection for Type Cultures (KCTC; Daejeon, Korea). For DNA extraction from Candida cell stocks, the yeasts were grown on yeast peptone dextrose (YPD) broth (Difco BD, Milan, Italy) at 180 rpm and 37 °C overnight. After cell counting with phase-contrast microscopy (40 × power) using a counting grid, DNA was extracted from the Candida cells (~2 × 10^8^ cells/mL) using the QIAamp UCP Pathogen Mini Kit according to the manufacturer’s manual.

### 2.3. Isolation of Candida Genomic DNA from Whole Blood

DNA extraction was performed using two different methods, the QIAamp UCP Pathogen Mini Kit (Qiagen, Hilden, Germany) and a newly developed simple boiling method using Chelex-100. First, DNA was extracted from the blood samples using the QIAamp UCP Pathogen Mini Kit and Lysing Matrix C tube (MP Biomedicals, Illkirch, France) according to the manufacturer’s instructions, and Candida DNA was eluted with 100 μL of elution buffer. Second, Candida DNA was extracted from the blood samples using the boiling method with Chelex-100 (Figure 1). Briefly, 200 µL of 2× red blood cell lysis buffer (Bio Basic, Toronto, ON, Canada) was added to 200 µL of whole blood sample. The sample was vortexed for 15 s first, followed by incubation at room temperature for 3 min and, later, centrifuged at 10,000 rpm for 5 min. After removing the supernatant, 500 µL of 10% Chelex-100 Resin (Bio-Rad Laboratories, Hercules, CA, USA) solution (10 mM Tris-HCl and 1 mM EDTA; pH 8.0) was added to the pellets. The suspension was boiled for 10 min at 100 °C in a heat block and then vortexed for 15 s three times. After filtration with a 3 µm Polycarbonate track-etched membrane filter (Whatman, Marlborough, Mass, USA), the supernatant was transferred to a new tube for subsequent experiments. The filtration process was performed using a SEPARA^®^ tube (GVS, Bologna, Italy). A 3 µm membrane was attached to the filtering unit using instant adhesive (UNITECH, Gyeonggi, Korea) after removing the existing filter (0.2 µm).

### 2.4. Primer Design

The *Candida* Pan LAMP primer set was designed within the conserved regions of partial ITS1, 5.8S rRNA gene and partial ITS2 of 6 Candida species (*C. albicans* MT640022.1_70-499, *C. glabrata* MT548912.1_350-885, *C. krusei* MZ507554.1_50-538, *C. tropicalis* LC639851.1_50-601, *C. parapsilosis* LC641867.1_130-786 and *C. auris* OL455790.1_1-300). For internal control, the LAMP primer set was designed within the conserved human glucose 6 phosphatase dehydrogenase (G6PD) gene. All LAMP primers, including two outer primers (forward primer F3 and backward primer B3), two inner primers (forward inner primer FIP and backward inner primer BIP) and two loop primers (forward loop primer FLP and backward loop primer BLP), were designed using Primer Explorer software (Version 4; Eiken Chemical Co., Tokyo, Japan). For the multiplex LAMP assay, a dye-labeled artificial nucleic acid + BLP sequence probe and a quencher-labeled displacement probe complementary to the artificial nucleic acid sequence were used. In this study, two types of artificial nucleic acids (35mers and 32mers) were used for multiplexing different fluorescence (Cy5 and Texas Red) quenched by BHQ2 and BHQ1, respectively. A Cy5-labeled 35-artificial oligomer-Ca Pan BLP was designed for Ca Pan BLP probe 1 and a Texas Red-labeled 32-artificial oligomer-internal control BLP was designed for internal control BLP probe 2. The quencher-labeled 35-oligonucleotide (BHQ2) or 30-oligonucleotide (BHQ1) were complementary to the artificial nucleic acid sequences of Ca Pan BLP probe 1 and internal control BLP probe 2, respectively. All primers were assessed for specificity before use in the LAMP assays via a BLAST search of sequences in GenBank (National Center for Biotechnology Information (NCBI), Bethesda, MD). All LAMP primers and probes were synthesized by Macrogen Inc. (Seoul, Korea; Table 1).

### 2.5. The Candida Pan/IC LAMP Assay

The *Candida* Pan/IC LAMP assay was performed using a Mmiso DNA amplification kit (Mmonitor, Deagu, South Korea). For the multiplex *Candida* Pan/IC LAMP assay, the reaction mixture was prepared with 12.5 μL of 2× reaction buffer, 1.25 μL of *Candida* Pan LAMP primer mix (20×), 0.3125 μL of internal control LAMP primer mix (20×), 1.25 μL of 9 μM quencher 1 solution for quenching the *Candida* Pan LAMP probe, 0.3125 μL of 9 μM quencher 2 solution for quenching the IC LAMP probe and 2 μL of sample DNA (with a final reaction volume of 25 μL). The composition of the *Candida* Pan LAMP primer mix (20×) included two outer primers at 4 µM (F3 and B3), two inner primers at 32 µM (FIP and BIP), 10 µM loopF primer (FLP), 4 µM loopB primer (BLP) and 6 µM loopB Cy5 probe. The composition of the internal control LAMP primer mix included two outer primers at 4 µM (F3 and B3), two inner primers at 32 µM (FIP and BIP), 10 µM loopF primer (FLP), 4 µM loopB primer (BLP) and 6 µM loopB Texas Red probe. The LAMP assay was run on a CFX 96 Touch Real-Time PCR Detection System (Bio-Rad Laboratories, Hercules, CA, USA) at 58 °C for 40 min. In the LAMP assay, negative controls (human blood DNA and distilled water) were used to set the baseline.

### 2.6. Real-Time PCR

To evaluate the performance of the *Candida* Pan/IC LAMP assay, real-time PCR was performed with *Candida* Pan [27] and *Candida* species real-time PCR primer sets [28,29] using the iQ Multiplex Powermix (Bio-Rad Laboratories, California, USA) on the CFX96 Touch Real-Time PCR Detection System (Bio-Rad Laboratories). The PCR cycling conditions of *Candida* Pan real-time PCR primer set were as follows: inactivation at 95 °C for 3 min, 39 cycles of denaturation at 95 °C for 15 s and annealing with fluorescence detection at 62 °C for 25 s. The PCR cycling conditions of the *Candida* species real-time PCR primer set were as follows: inactivation at 50 °C for 2 min and 95 °C for 10 min, 39 cycles of denaturation at 95 °C for 15 s and annealing with fluorescence detection at 60 °C for 1 min.

### 2.7. Limit of Detection (LOD) Tests

The LOD of the *Candida* Pan/IC LAMP was determined using six *Candida* spp., including *C. albicans, C. krusei, C. tropicalis, C. parapsilosis, C. auris* and *C. glabrata*. *Candida* DNA (1.0 × 10^7^ cells) was serially diluted 10-fold, from 1.0 × 10^7^ cells/μL to 1.0 × 10^0^ cells/μL and used to determine the LOD of the multiplex Candia Pan/IC LAMP assay. In addition, the LOD of the *Candida* Pan/IC LAMP was tested with serially diluted blood samples spiked with *C. albicans* (from 10^7^ to 10^0^). The LOD of the *Candida* Pan/IC LAMP assay was compared with that of the conventional *Candida* Pan RT-PCR. All tests were repeated three times and determined as the minimum concentration in a 10-fold dilution series, at which three of three replicates were amplified.

### 2.8. Statistical Analysis

The confidence intervals (CI) for sensitivity and specificity were set at 95%. The sensitivity, specificity and 95% CI for the assays were calculated using a diagnostic test evaluation calculator program (https://www.medcalc.org/calc/diagnostic_test.php, accessed on 21 December 2021).

## 3. Results

### 3.1. Optimization of the Chelex-100/Boiling Method for Nucleic Acid Extraction

To optimize the Chelex-100/boiling method, different concentrations of the Chelex 100 Resin solutions (0%, 5% and 10%) were tested using *C. albicans* real-time PCR and the *Candida* Pan/IC LAMP assay for *C. albicans* DNA extracted from the whole blood samples spiked with *Candida* cells (total cell concentration of 10^7^/mL) (Figure 2A). For DNA extraction with 0%, 5% and 10% Chelex-100 resin solutions, real-time PCR and the *Candida* Pan/IC LAMP assay showed Ct 28.84/26.16/25.05 and Ct 13.59/13.49/12.75, respectively. Thus, the 10% Chelex-100 resin solution was determined to be the optimum concentration of Chelex-100 for the Chelex-100/boiling method. Next, the performance of the Chelex-100/boiling method was compared with that of the commercial QIAamp UCP Pathogen Mini Kit (Qiagen, Hilden, Germany) against whole blood samples spiked with *C. albicans* using *C. albicans* qPCR and *Candida* Pan/IC LAMP assay (Figure 2B, Table 2). As a result, the detection limits of the *C. albicans* qPCR primer set were 10^5^/mL and 10^4^/mL in DNA samples extracted using the Chelex-100/boiling and QIAamp UCP Pathogen Mini Kit, respectively. Interestingly, the *Candida* Pan/IC LAMP assay showed lower detection limits (10^4^/mL and 10^3^/mL) than those obtained by qPCR for DNA extracted using the Chelex-100/boiling and QIAamp UCP Pathogen Mini Kit, respectively. Although the Chelex-100/boiling method showed lower efficiency than the commercial QIAamp UCP Pathogen Mini Kit, *Candida* Pan/IC LAMP, followed by Chelex-100/boiling DNA extraction, showed similar results to qPCR for DNA samples extracted using the QIAamp UCP Pathogen Mini Kit. These results suggest that the *Candida* Pan/IC LAMP with Chelex-100/boiling DNA extraction is useful for the rapid diagnosis of candidemia.

### 3.2. Comparison of Detection Limits of the Candida Pan/IC LAMP Assay and Two qPCR (Pan and Candida Species) against Candida Species

To confirm the performance of the *Candida* Pan/IC LAMP assay, the detection limit of the LAMP assay was compared with that of *Candida* Pan/*Candida* species qPCRs for six *Candida* species, including *C. albicans, C. glabrata, C. tropicalis, C. krusei, C. parapsilosis* and *C. auris.* DNA samples from all *Candida* species were extracted using the QIAamp UCP Pathogen Mini Kit (Table 3). For *C. albicans and C. krusei,* the *Candida* Pan/IC LAMP assay showed the lowest detection limit (10^3^) among the three tested assays. The detection limits of *Candida* Pan qPCR and *Candida* species qPCR (*C. albicans* and *C. krusei*) [28] were 10^5^/10^6^ and 10^4^/10^5^ for *C. albicans and C. krusei,* respectively. For *C. glabrata* and *C. tropicalis,* the *Candida* Pan/IC LAMP assay and *Candida* species qPCR (*C. glabrata* and *C. tropicalis*) [28] showed the same detection limit (10^5^). The detection limit of *Candida* Pan qPCR was 10^6^/10^6^ for *C. glabrata* and *C. tropicalis*, respectively. qPCR specific for *C**. parapsilosis* [28] and *C. auris* [29] showed the lowest detection limit (10^4^) among the three tested assays. The detection limits of *Candida* Pan qPCR were 10^6^ and 10^7^ for *C. parapsilosis* and *C. auris*, respectively. The *Candida* Pan/IC LAMP assay showed the same detection limits (10^5^) for *C*. *parapsilosis* and *C. auris*.

### 3.3. Sensitivity and Specificity of the Candida Pan/IC LAMP Assay with Two qPCR Assays against Candida Clinical Sample DNA Extracted by QIAamp UCP Pathogen Mini Kit and Chelex-100/Boiling Method

To confirm the clinical performance of the Chelex-100/boiling DNA extraction method and the *Candida* Pan/IC LAMP assay, DNA extraction was performed using two different methods, the QIAamp UCP Pathogen Mini Kit and Chelex-100/boiling method; the sensitivities of the *Candida* Pan/IC LAMP assay were compared with those of the *Candida* Pan qPCR and specific *Candida* species qPCR for 36 clinical samples from patients infected with *C. albicans* (9), *C. glabrata* (9), *C. tropicalis* (9) and *C. parapsilosis* (9) (Table 4). Within 40 min, the *Candida* Pan/IC LAMP assay showed 100% sensitivity for two kinds of *Candida* clinical sample DNA extracted by QIAamp UCP Pathogen Mini Kit and Chelex-100/boiling method, respectively. Conventional *Candida* Pan and *Candida* species qPCR showed the same sensitivity (86.11%) for 36 *Candida* clinical sample DNA extracted using the QIAamp UCP Pathogen Mini Kit. However, for *Candida* clinical sample DNA extracted by the Chelex-100/boiling method, the *Candida* Pan qPCR and *Candida* species qPCR showed 22% and 44% sensitivity, respectively. For 100 negative clinical samples (non-infected), the specificity of the three assays was 100% (Table 4).

### 3.4. Cross-Reactivity Test

To confirm the possibility of cross-reactivity of the *Candida* Pan/IC LAMP assay, the *Candida* Pan/IC LAMP assay was tested with other bacterial samples, including *Escherichia coli, Enterococcus faecium, Klebsiella spp., Staphylococcus aureus* and *Staphylococcus epidermidis* samples (Table 5). The *Candida* Pan/IC LAMP assay showed no cross-reactivity with other fungal infection samples, suggesting that the LAMP assay can specifically detect *Candida* species.

## 4. Discussion

Invasive candidiasis (IC) is a serious cause of morbidity and mortality [26,30]. In the hospital, *Candida* spp. account for 8–9% of all nosocomial bloodstream infections and the risk is higher in intensive care unit (ICU) patients and cancer patients [31,32]. Candidemia has an associated mortality rate of up to 25% and a fast diagnosis followed by early adequate antifungal therapy can significantly reduce premature mortality [33,34].

In this study, we developed a Chelex-100/boiling DNA extraction method (within 20 min) and *Candida* Pan/IC LAMP assay (40 min) for the rapid diagnosis of candidemia. The Chelex-100/boiling method showed a slightly lower efficiency than the commercial QIAamp UCP Pathogen Mini Kit; however, the Chelex-100/boiling DNA extraction method (within 20 min) extracted DNA 3–4 times faster than commercial QIAamp UCP Pathogen Mini kits (60–90 min). In addition, Chelex-100/boiling DNA extraction followed by the *Candida* Pan/IC LAMP assay showed similar results to the reference qPCR for DNA samples extracted using the QIAamp UCP Pathogen Mini Kit. Interestingly, the LAMP assay showed superior performance (100% sensitivity) compared with the conventional reference qPCRs (86.11% sensitivity) against 36 numbers of *Candida* clinical DNA samples extracted by both the Chelex-100/boiling method and commercial QIAamp UCP Pathogen Mini Kit. In addition, the *Candida* Pan/IC LAMP assay showed 100% specificity against 100 non-infected clinical samples.

As fungal cell walls are difficult to break with conventional extraction methods, obtaining DNA from fungi is more difficult than extracting nucleic acids from bacteria or mammalian cells [35]. For this reason, nucleic acid extraction methods to detect fungus-infected blood generally require additional procedures (e.g., mechanical, enzymatic and/or chemical methods) to disrupt the fungal cell wall [36]. Therefore, these nucleic acid extraction methods consist of a complicated procedure and it takes a long time to extract the nucleic acid [37].

Nucleic acid extraction using cheliex-100 is known as a fast and easy method for nucleic acid extraction from various samples, such as forensics, blood, parasites, virus and bacteria [38,39]. The first protocol for DNA extraction using Chelex-100 was developed by Walsh et al. [40]. Chelex-100 has been mainly used in forensics in conjunction with thermal denaturation to extract nucleic acids from trace cells or blood. Boiling the sample not only releases the DNA from the cells into the solution, but also promotes the binding of Chelex-100 to the magnesium ion, which is a cofactor of deoxyribonuclease. DNA degradation is prevented because the binding of Chelex-100 to magnesium ions results in the inactivation of deoxyribonuclease. In this study, after removing red blood cells from the blood using a red blood cell lysis solution, cells that were not lysed were separated by centrifugation. After heating the separated cells using a Chelex-100 solution, the nucleic acids were separated using a 3 µm filter. Additionally, a bead beating step was added to the Chelex-100/boiling method to further increase *Candida* cell destruction. However, there was no significant difference in the ct value of the LAMP assay, so this step was excluded (Appendix A).

The *Candida* Pan LAMP primer set was designed using Primer Explorer software by multialigning conserved regions of partial ITS1, 5.8S rRNA gene and partial ITS2 genes of *C. albicans, C. glabrata, C. tropicalis, C. krusei, C. parapsilosis* and *C. auris*. However, it was not possible to design the BLP primer because B1 and B2 of the designed primer set were too close. The LOOP primer explosively amplified the LAMP reaction. In this study, for the development of the rapid *Candida* Pan LAMP primer set, an artificial sequence (TTCGCTGCGCTCTTCA) capable of reacting with the LOOP primer was added between B1 and B2 to design the BIP primer. Indeed, the *Candida* Pan LAMP primer set with the BLP primer of the same artificial sequence showed an increased reaction rate compared to the primer set without BLP (Appendix A). In addition, we used this BLP + Cy5-labeled 35-artificial oligomer as a Candida Pan probe; since the BLP of this artificial sequence is not involved in the Candida target gene, it only responds to the operation of BIP without a non-specific reaction.

Our study has a limitation. The Candida Pan/IC LAMP assay was performed with a relatively small sample size of positive Candida clinical samples (36), which resulted in widened confidence intervals for our estimates of diagnostic accuracy. However, considering that the Candida Pan/IC LAMP assay showed higher sensitivity than the two conventional PCR and 100% specificity to the negative samples, the Candida Pan/IC LAMP assay is sufficiently competitive for commercial development.

## 5. Conclusions

In this study, we developed a fast candidemia detection system including Chelex-100/boiling DNA extraction and the *Candida* Pan/IC LAMP assay, which is capable of diagnosing *Candida* species in blood within 1 h. In a sensitivity test with *Candida* clinical samples, the *Candida* Pan/IC LAMP assay showed superior performance to the two reference qPCRs. Thus, Chelex-100/boiling DNA extraction followed by the *Candida* Pan/IC LAMP assay could serve as a useful fast molecular diagnostic test for *Candida* spp. in blood.

## Figures and Tables

**Figure 1 pathogens-11-00111-f001:**
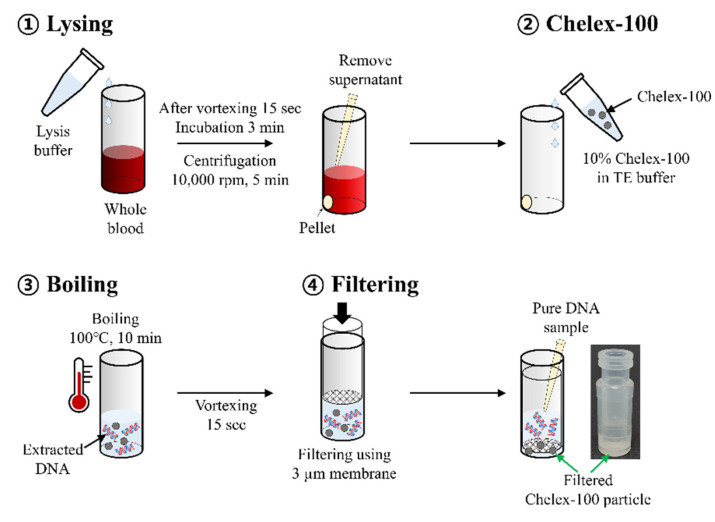
Schematic of Chelex-100/boiling DNA extraction method.

**Figure 2 pathogens-11-00111-f002:**
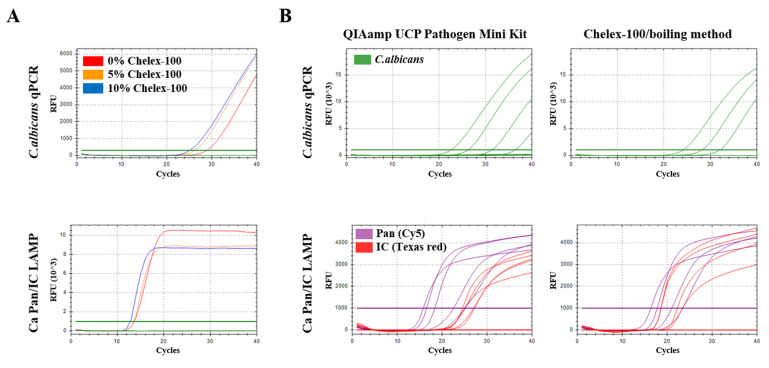
Optimization of the Chelex-100/boiling DNA extraction method. (**A**) Different concentration ratios of Chelex-100 (0%, 5% and 10%). (**B**) Comparison of detection limit of *Candida albicans* qPCR and *Candida* Pan/IC LAMP assay on two kinds of 10-fold serial diluted DNA samples extracted by Chelex-100/boiling and QIAamp UCP Pathogen Mini Kit, respectively.

**Table 1 pathogens-11-00111-t001:** The LAMP primer sets and qPCR primer sets used in the study.

Target	Name	Sequence (5′-3′)	Length (mer)	Reference
*Candida* Pan (Ca Pan, partial ITS1, 5.8S rRNA gene and partial ITS2)	Ca Pan F3	AAA ACT TTC AAC GGA T	19	Present study
	Ca Pan B3	ACG CTC AAA CAG GCA	15	
	Ca Pan FIP	CAA KTC ARA YTA WKT ATC GCA STT CCT CTT GGT TCT CGC ATC G	43	
	Ca Pan BIP	CGT GAA TCA TCG AAR YYT TT TTC GCT GCG CTC TTC ATT GGC GCA ATG TGC GT	53	
	Ca Pan FLP	ACG TAT CGC ATT TCG CTG C	19	
	Ca Pan BLP	TTC GCT GCG CTC TTC A	16	
	Ca Pan BLP_CY5 probe1	[CY5]-GTC AGT GCA GGC TCC CGT GTT AGG ACG AGG GTA GGT TCG CTG CGC TCT TCA	51	
Internal control (IC, G6PD)	IC G6PD F3	TGT CAC CAG CAA CAT CTC GA	20	Present study
	IC G6PD B3	TCC TCA GGG AAG CAA ATG AC	18	
	IC G6PD FIP	ATA GCA GAG AGG CTG CCT ACG GTT TTG ATG TCC CCT GTC CCA	45	
	IC G6PD BIP	AAG AAA AGC AGA CGC AGC TTT TTG GGG CTG TTT GCG GAT T	43	
	IC G6PD FLP	GGG GTG GCC ATG GAG TGC	18	
	IC G6PD BLP	TCC CAA CCT CAA TGC CCT GC	20	
	IC G6PD BLP TEX probe 2	[Texas red] –CGG GCC CGT ACA AAG GGA ACA CCC ACA CTC CGT CCC AAC CTC AAT GCC CTG C	52	
Quencher probe 1		CCT ACC CTC GTC CTA ACA CGG GAG CCT GCA CTG AC-BHQ2	35	
Quencher probe 2		GAG TGT GGG TGT TCC CTT TGT ACG GGC CCG-BHQ1	30	
*Candida* Pan (Ca Pan) RT-PCR	CP PCR F	CCT GTTT GAG CGT CRT TT	17	[27]
	CP PCR R	TCC GCT TAT TGA TAT	18	
*C. albicans* (CA) RT-PCR	CA PCR F	CTT GGT ATT TTG CAT GTT GCT CTC	24	[28]
	CA PCR R	GTC AGA GGC TAT AAC ACA CAG CAG	24	
	CA PCR probe	[FAM] - TTT ACC GGG CCA GCA TCG GTT T – BHQ1	22	
*C. glabrata* (CG) RT-PCR	CF PCR F	GCG CCC CTT GCC TCT C	16	[28]
	CF PCR R	CCC AGG GCT ATA ACA CTC TAC ACC	24	
	CF PCR probe	[HEX] – TGG GCT TGG GAC TCT CGC AGC – BHQ1	21	
*C. tropicalis* (CT) RT-PCR	CT PCR F	GCG GTA GGA GAA TTG CGT T	19	[28]
	CT PCR R	TCA TTA TGC CAA CAT CCT AGG TTT A	25	
	CT PCR probe	[CY5] – CGC AGT CCT CAG TCT AGG CTG GCA G – BHQ2	25	
*C. krusei* (CK) RT-PCR	CK PCR F	CTCA GAT TTG AAA TCG TGC TTT G	23	[28]
	CK PCR R	GGG GCT CTC ACC CTC CTG	18	
	CK PCR probe	[TEX] – CAC GAG TTG TAG ATT GCA GGT TGG AGT CTG – BHQ1	30	
*C. parapsilosis* (CP) RT-PCR	CP PCR F	GAT CAG ACT TGG TAT TTT GTA TGT TAC TCT C	31	[28]
	CP PCR R	CAG AGC CAC ATT TCT TTG CAC	21	
	CP PCR probe	[FAM] – CCT CTA CAG TTT ACC GGG CCA GCA TCA – BHQ1	27	
*C. auris* (CR) RT-PCR	CR PCR F	CGT GAT GTC TTC TCA CCA ATC T	22	[29]
	CR PCR R	TAC CTG ATT TGA GGC GAC AAC	21	

**Table 2 pathogens-11-00111-t002:** Limit of detection (LOD) tests of *Candida* species qPCRs and the *Candida* Pan/IC LAMP assay for two *Candida*-spiked blood DNA samples extracted by QIAamp UCP Pathogen Mini Kit and Chelex-100/boiling method.

DNA Extraction Method	PCR Analysis	Primer Sets	Total Concentration (cells/mL)							
			10^7^	10^6^	10^5^	10^4^	10^3^	10^2^	10^1^	DW *
QIAamp UCP Pathogen Mini Kit	qPCR	*C. albicans*	22.48	26.15	31.29	36.27	N/A	N/A	N/A	N/A
	Multiplex RT LAMP	Cy5 (c. pan)	15.80	16.53	18.35	22.54	25.14	N/A	N/A	N/A
		Tex (IC)	25.01	27.20	27.01	24.53	25.10	N/A	N/A	N/A
Chelex-100/boiling	qPCR	*C. albicans*	24.12	28.11	31.99	N/A	N/A	N/A	N/A	N/A
	Multiplex RT LAMP	Cy5 (c. pan)	16.36	17.78	21.04	23.43	N/A	N/A	N/A	N/A
		Tex (IC)	23.43	22.25	18.63	18.59	N/A	N/A	N/A	N/A

* DW, distilled water; N/A, not available.

**Table 3 pathogens-11-00111-t003:** Limit of detection (LOD) tests of Pan/*Candida* species qPCRs and the *Candida* Pan/IC LAMP assay for 6 *Candida* spp.

*Candida* Species	Primer Sets		Total Concentration (cells/mL)							
			10^7^	10^6^	10^5^	10^4^	10^3^	10^2^	10^1^	DW *
*C. albicans*	qPCR	*Candida* Pan	24.17	32.71	36.66	N/A	N/A	N/A	N/A	N/A
		*C. albicans*	22.50	26.60	31.35	39.78	N/A	N/A	N/A	N/A
	LAMP	Cy5 (c. pan)	15.20	16.75	22.45	27.44	28.01	N/A	N/A	N/A
		Tex (IC)	N/A	N/A	N/A	N/A	N/A	N/A	N/A	N/A
*C. glabrata*	qPCR	*Candida* Pan	22.02	31.08	N/A	N/A	N/A	N/A	N/A	N/A
		*C. glabrata*	21.07	28.25	35.50	N/A	N/A	N/A	N/A	N/A
	LAMP	Cy5 (c. pan)	15.04	17.4.	23.76	N/A	N/A	N/A	N/A	N/A
		Tex (IC)	N/A	N/A	N/A	N/A	N/A	N/A	N/A	N/A
*C. tropicalis*	qPCR	*Candida* Pan	22.27	30.38	N/A	N/A	N/A	N/A	N/A	N/A
		*C. tropicalis*	23.69	29.22	36.38	N/A	N/A	N/A	N/A	N/A
	LAMP	Cy5 (c. pan)	15.76	18.36	26.04	N/A	N/A	N/A	N/A	N/A
		Tex (IC)	N/A	N/A	N/A	N/A	N/A	N/A	N/A	N/A
*C. krusei*	qPCR	*Candida* Pan	20.60	28.44	N/A	N/A	N/A	N/A	N/A	N/A
		*C. krusei*	22.46	28.42	38.84	N/A	N/A	N/A	N/A	N/A
	LAMP	Cy5 (c. pan)	17.00	19.31	25.34	34.45	N/A	N/A	N/A	N/A
		Tex (IC)	N/A	N/A	N/A	N/A	N/A	N/A	N/A	N/A
*C. parapsilosis*	qPCR	*Candida* Pan	20.38	24.85	34.73	N/A	N/A	N/A	N/A	N/A
		*C. parapsilosis*	21.34	25.80	31.48	36.97	N/A	N/A	N/A	N/A
	LAMP	Cy5 (c. pan)	17.80	20.34	28.92	N/A	N/A	N/A	N/A	N/A
		Tex (IC)	N/A	N/A	N/A	N/A	N/A	N/A	N/A	N/A
*C. auris*	qPCR	*Candida* Pan	35.51	N/A	N/A	N/A	N/A	N/A	N/A	N/A
		*C. auris*	21.59	25.14	30.21	36.63	N/A	N/A	N/A	N/A
	LAMP	Cy5 (c. pan)	15.97	17.54	21.58	N/A	N/A	N/A	N/A	N/A
		Tex (IC)	N/A	N/A	N/A	N/A	N/A	N/A	N/A	N/A

* DW, distilled water; N/A, not available.

**Table 4 pathogens-11-00111-t004:** Comparison of sensitivity and specificity of multiplex *Candida* Pan/IC LAMP assay with reference pan and mono *Candida* species RT-PCR against *Candida* and non-infectious clinical samples.

Clinical Samples		QIAamp UCP Pathogen Mini Kit				Boiling and Filtering Method			
		qPCR		Multiplex LAMP		qPCR		Multiplex LAMP	
		*Candida* pan	*Candida* Species	Cy5 (C. pan)	Tex (IC)	*Candida* pan	*Candida* Species	Cy5 (C. pan)	Tex (IC)
*Candida Spp.* (n = 36)	P/N	31/5	31/5	36/0	27/9	8/28	16/20	36/0	28/8
	Sensitivity (95% CI-	86.11% [70.50–95.33]	86.11% [70.50–95.33]	100% [90.26–100.00]	75.00% [57.80–87.88]	22.22% [10.12–39.15]	44.44% [27.94–61.90]	100% [90.26–100.00]	77.77% [60.85–89.88]
*C. albicans* (n = 9)	P/N	8/1	8/1	9/0	6/3	1/8	5/4	9/0	9/0
	Sensitivity	88.89%	88.89%	100%	66.67%	11.11%	55.56%	100%	100%
*C. glabrata* (n = 9)	P/N	8/1	8/1	9/0	7/2	2/7	3/6	9/0	6/3
	Sensitivity	88.89%	88.89%	100%	77.78%	22.22%	33.33%	100%	66.67%
*C. tropicalis* (n = 9)	P/N	6/3	6/3	9/0	8/1	2/7	3/6	9/0	6/3
	Sensitivity	66.67%	66.67%	100%	88.89%	22.22%	33.33%	100%	66.67%
*C. parapsilosis* (n = 9)	P/N	9/0	9/0	9/0	6/3	3/6	5/4	9/0	7/2
	Sensitivity	100%	100%	100%	66.67%	33.33%	55.56%	100%	77.78%
Non-infection (n = 100)	P/N	0/100	0/100	0/100	100/0	0/100	0/100	0/100	100/0
	Sensitivity (95% CI)	N/A	N/A	N/A	100% [96.38–100.00]	N/A	N/A	N/A	100% [96.38–100.00]
	Specificity (95% CI)	100% [96.38–100.00]	100% [96.38–100.00]	100% [96.38–100.00]	N/A	100% [96.38–100.00]	100% [96.38–100.00]	100% [96.38–100.00]	N/A

**Table 5 pathogens-11-00111-t005:** Cross-reactivity of the *Candida* Pan/IC LAMP against other bacterial infection samples.

	*Candida* Pan/IC LAMP			
	Cy5 (*Candida* pan)		Texas Red (Internal Control)	
Samples	Ct	RFU *	Ct	RFU *
*Escherichia coli*	N/A	19.1	N/A	23.3
*Enterococcus faecium*	N/A	31.1	N/A	22.6
*Klebsiella spp.*	N/A	20.0	N/A	28.3
*Staphylococcus aureus*	N/A	44.0	N/A	39.0
*Staphylococcus epidermidis*	N/A	57.7	N/A	42.6
Human whole blood DNA	N/A	−17.9	29.68	4186
Distilled water	N/A	1.76	N/A	−0.889

* RFU, relative fluorescence units.

## Data Availability

Data are contained within the article.

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
