# Peer review of "Development of a Simple DNA Extraction Method and Candida Pan Loop-Mediated Isothermal Amplification Assay for Diagnosis of Candidemia"

_pathogens, 2022, doi:10.3390/pathogens11020111_

Round 1

Reviewer 1 Report

The authors describe in the manuscript the development of a novel multiplex LAMP assay for the detection of Candida bloodstream infections. A rapid isolation method of Candida DNA from blood samples was also developed. The study is well-designed and well-conducted. The manuscript is carefully prepared and all its sections are clearly described.

I have a few minor comments that need to be considered/corrected:

  1. Line 39-40: Please rewrite the following sentence: “Therefore, various non-culture-based diagnostic methods such as mannan, (1-3)-β-d-glucan, and PCR have been developed”. (I suppose it should be: “…mannan, (1-3)-β-d-glucan assay, (1-3)-β-d-glucan assay…”).
  2. Line 57: Please rewrite the following sentence: “..and then robustly amplify the target gene at 58–65°C using Bst polymerase..” (not only Bst polymerase could be used in the LAMP assay).
  3. Section 2.4. Primer design: Could you please provide sequence accession numbers of genes that were used to design the LAMP primers?
  4. Line 151: Please correct the following sentence: “…and 2 μL of sample RNA…” (I suppose it should be: “…and 2 μL of sample DNA…”).
  5. Tables 2-4: Please explain the abbreviations (N/A; DW).
  6. Table 4: In the caption of the table, you write about “DNA samples extracted from Candida-spiked whole blood”, while section 3.4. is about clinical samples. Please clarify this issue.
  7. Discussion section: Please compare the time needed to isolate DNA from a sample using a developed protocol and a commercial kit (QIAamp UCP Pathogen Mini Kit).

Author Response

Response to Reviewer 1 Comments

We would like to thank you for your interest and detailed comments on our manuscript. We really appreciate it. We revised our manuscript according to the comments.

Reviewer 1 comment

The authors describe in the manuscript the development of a novel multiplex LAMP assay for the detection of Candida bloodstream infections. A rapid isolation method of Candida DNA from blood samples was also developed. The study is well-designed and well-conducted. The manuscript is carefully prepared and all its sections are clearly described.

I have a few minor comments that need to be considered/corrected:

Point 1: Line 39-40: Please rewrite the following sentence: “Therefore, various non-culture-based diagnostic methods such as mannan, (1-3)-β-d-glucan, and PCR have been developed”. (I suppose it should be: “…mannan, (1-3)-β-d-glucan assay, (1-3)-β-d-glucan assay…”).

Response 1: We changed the sentence based on the comment following as “Therefore, various non-culture-based diagnostic methods such as immunoassay (mannan, anti-mannan antibodies and (1-3)-β-d-glucan (BDG) assay) [9-11] and PCR [8]  have been developed; however, detection methods using mannan, anti-mannan antibodies and BDG have been reported to have low specificity [10,12] and PCR-based diagnosis is time-consuming although the detection specificity is high.” (Line 39-43)

Point 2: Line 57: Please rewrite the following sentence: “..and then robustly amplify the target gene at 58–65°C using Bst polymerase..” (not only Bst polymerase could be used in the LAMP assay).

Response 2:  We changed the sentence based on the comment following as “and then robustly amplify the target gene at 58–65°C using Bst or Bsm polymerases with DNA-strand displacement activity [18, 19]. (Line 66-68)

Point 3: Section 2.4. Primer design: Could you please provide sequence accession numbers of genes that were used to design the LAMP primers?

Response 3:  We added the sequence accession numbers of genes in section 2.4. Primer design based on the comment following as “The Candida Pan LAMP primer set was designed within the conserved regions of partial ITS1, 5.8S rRNA gene, and partial ITS2 of 6 Candida species (C. albicans MT640022.1_70-499, C. glabrata MT548912.1_350-885, C. krusei MZ507554.1_50-538, C. tropicalis LC639851.1_50-601, C. parapsilosis LC641867.1_130-786 and C. auris OL455790.1_1-300).” (Line 142-146)

Point 4: Line 151: Please correct the following sentence: “…and 2 μL of sample RNA…” (I suppose it should be: “…and 2 μL of sample DNA…”).

Response 4:  As your comment, we changed it. (RNA à DNA) (Line 174)

Point 5: Tables 2-4: Please explain the abbreviations (N/A; DW).

Response 5:  As your comment, we added the abbreviations (N/A: not available; DW: Distilled Water) in Table 2-3 and changed – to N/A in Table 4.

Point 6: Table 4: In the caption of the table, you write about “DNA samples extracted from Candida-spiked whole blood”, while section 3.4. is about clinical samples. Please clarify this issue.

Response 6:  We have a mistake. We changed the caption of the table4 based on the comment following as “Comparison of sensitivity and specificity of multiplex Candida Pan/IC LAMP assay with reference pan and mono Candida species RT-PCR against Candida and non-infectious clinical samples.” (Line 284-285)

Point 7: Discussion section: Please compare the time needed to isolate DNA from a sample using a developed protocol and a commercial kit (QIAamp UCP Pathogen Mini Kit).

Response 7:  We added the sequence based on the comment following as “however, the Chelex-100/boiling DNA extraction method (within 20 min) extracted DNA 3-4 times faster than commercial QIAamp UCP Pathogen Mini kits (60-90 min). In addition,” (Line 325-327)

Reviewer 2 Report

The authors stated an important problem and its solution in an eloquent manner. The manuscript is well written and description of all aspects are well presented. Some editing and additional details are required. These are listed below:

A. Introduction:

1. Line 41 -  Use the BGD acronym before.

2. Line 41 - How much low the specificity is? Please use reference.

3. Line 57 - Add "strand" before a.

B. Materials and Methods:

1. Line 77 - Please mention "Whole blood" for blood samples.

2. Line 77-78 - 

Please provide details about Clinical samples:

i. How were the healthy volunteers selected and determined?
ii. How were the Candida infected patients been tested?

3. Line 107 - What is RT?

4. Line 160 - How the contamination issue and cross reaction issue was handled? How the specificity was determined? 

5. Line 183 - How the positive and negative case was determined? What criteria was been used? 

6. Statistical analysis section is missing. Please add that. 

C. Discussion:

  1. Line 284 - Remove "-". from mammalian.
  2. Line 290 - Remove "-". from nucleic.
  3. What are the limitations of the conducted study? 
  4.  

Author Response

Response to Reviewer 2 Comments

We would like to thank you for your interest and detailed comments on our manuscript. We really appreciate it. We revised our manuscript according to the comments.

Reviewer 2

The authors stated an important problem and its solution in an eloquent manner. The manuscript is well written and description of all aspects are well presented. Some editing and additional details are required. These are listed below:

  1. Introduction:
  2. Line 41 - Use the BGD acronym before.

Response A1: We added it based on the comment following as “(1-3)-β-d-glucan (BDG) assay” (Line 40)

  1. Line 41 - How much low the specificity is? Please use reference.

Response A2: We added the reference. (Line 42)

  1. Line 57 - Add "strand" before a.

Response A3: We changed the sentence based on the comment following as “and then robustly amplify the target gene at 58–65°C using Bst or Bsm polymerases with DNA-strand displacement activity [18, 19]” (Line 66-68)

  1. Materials and Methods:
  2. Line 77 - Please mention "Whole blood" for blood samples.

Response B1: We added it based on the comment following as “blood samples à Whole blood samples” (line 88)

  1. Line 77-78 - Please provide details about Clinical samples:
  2. How were the healthy volunteers selected and determined?
  3. How were the Candida infected patients been tested?

Response B2:

  1. response: We have a mistake. It is not healthy volunteers. Thus, we changed it following as “A total of 136 clinical whole blood samples were collected from Candida-infected (n = 36) and non-infected patients (normal control, n=100) at Korea University Guro Hospital from January 2019 to August 2021.” (Line 88-90)
  2. response: It was confirmed by VITEK 2 COMPACT system. Thus, we added it in Materials and Methods section following as “Candida clinical samples were confirmed by VITEK 2 COMPACT system (bioMérieux, Durham, NC) using VITEK®2 YST ID card (bioMérieux, Durham, NC).” (Line 90-92)

  1. Line 107 - What is RT?

Response B3: We changed it based on your comments following as “RTà room temperature”. (Line 127)

  1. Line 160 - How the contamination issue and cross reaction issue was handled? How the specificity was determined?

Response B4: To minimize false positives due to amplicon aerosol contamination, preparation of the LAMP reaction mixture, loading of template DNA, incubation and measurement of the reaction mixture were performed in a separate laboratory space. The sensitivity, specificity and 95% CI for the assays were calculated using a diagnostic test evaluation calculator program (https://www.medcalc.org/calc/diagnostic_test.php). (Line210-213)

  1. Line 183 - How the positive and negative case was determined? What criteria was been used?

Response B5: In the three LOD tests, if the first showed a detection limit of 10^2, the second and third showed a detection limit of 10^3, and 10^4, the detection limit of this test was determined to be 10^4. FDA EUA guidelines for COVID-19 diagnostic tests were used with slight modifications.

  1. Statistical analysis section is missing. Please add that.

Response B6: We added the Statistical analysis in Materials & Methods section and Table 4 based on the comment following as “Confidence intervals (CI) for sensitivity and specificity were set at 95%. The sensitivity, specificity and 95% CI for the assays were calculated using a diagnostic test evaluation calculator program (https://www.medcalc.org/calc/diagnostic_test.php).” (Line 210-213)

  1. Discussion:
  2. Line 284 - Remove "-". from mammalian.

Response C1: We removed it based on your comments.

  1. Line 290 - Remove "-". from nucleic.

Response C2: We removed it based on your comments.

  1. What are the limitations of the conducted study?

Response C3: We added the sentence based on the comment following as “Our study has a limitation. The Candida Pan/IC LAMP assay was performed with a relatively small sample size of positive Candida (36) clinical samples, which resulted in widened confidence intervals for our estimates of diagnostic accuracy. However, considering that the Candida Pan/IC LAMP assay showed the higher sensitivity compared to the two conventional PCR, and 100% specificity to the negative samples, the Candida Pan/IC LAMP assay is sufficiently competitive in commercial development.” (Line 369-376)
